# Optogenetic and Chemogenic Control of Pain Signaling: Molecular Markers

**DOI:** 10.3390/ijms241210220

**Published:** 2023-06-16

**Authors:** Josue Vidal Espinosa-Juárez, Erwin Chiquete, Bruno Estañol, José de Jesús Aceves

**Affiliations:** 1Escuela de Ciencias Químicas Sede Ocozocoautla, Universidad Autónoma de Chiapas, Ocozocoautla de Espinosa 29140, Mexico; josue.espinosa@unach.mx; 2Instituto Nacional de Ciencias Médicas y Nutrición Salvador Zubirán, Mexico City 14080, Mexicobruno.estanolv@incmnsz.mx (B.E.)

**Keywords:** pain, VGluT2, optomodulation, chemogenetic

## Abstract

Pain is a complex experience that involves physical, emotional, and cognitive aspects. This review focuses specifically on the physiological processes underlying pain perception, with a particular emphasis on the various types of sensory neurons involved in transmitting pain signals to the central nervous system. Recent advances in techniques like optogenetics and chemogenetics have allowed researchers to selectively activate or inactivate specific neuronal circuits, offering a promising avenue for developing more effective pain management strategies. The article delves into the molecular targets of different types of sensory fibers such as channels, for example, TRPV1 in C-peptidergic fiber, TRPA1 in C-non-peptidergic receptors expressed differentially as MOR and DOR, and transcription factors, and their colocalization with the vesicular transporter of glutamate, which enable researchers to identify specific subtypes of neurons within the pain pathway and allows for selective transfection and expression of opsins to modulate their activity.

## 1. Introduction

Pain is a complex experience that involves not only the physical sensation of tissue damage or injury, but also emotional and cognitive components [1]. It is important to distinguish between pain and nociceptive behavior, which refers to the body’s automatic and reflexive response to tissue damage or injury. While nociceptive behavior helps protect the body from further injury and promotes healing, pain is a subjective experience that is influenced by a variety of factors, including emotions, past experiences, and cognitive processes [2]. Understanding the differences between pain and nociceptive behavior is crucial because those events occur at different times during the pathophysiological processes and will require developing complementary therapeutics strategies [3].

The physiological process for pain perception is generated through the transmission of information from the site of damage to the central nervous system (CNS). The neurons that capture signals are essential elements in the perception of pain. These terminal endings confront a pseudo unipolar neuron that is in the dorsal root ganglion. The axon of this neuron is divided into two branches, one projecting to the periphery and the other to the CNS [4,5]. All sensory information is captured by the primary afferent fibers of various types, which carry the signal to the dorsal horn of the spinal cord. The sensory primary afferents are categorized into three main groups, according to their diameter, degree of myelination, and driving speed. Neurons whose cell body is large in diameter are A-β neurons; they have myelinated axons, which gives them the characteristic of fast conduction. The medium-sized neurons are called A-δ and have lightly myelinated axons and moderate conduction velocities, while neurons with small diameter are C fibers, whose axon is unmyelinated, and their conduction velocity is the slowest [6].

With recent techniques such as optogenetics or chemogenetics it is possible to activate or inactivate neuronal circuits in a specific way [7,8]; thus, it is essential to identify molecular targets from each fiber, such as channels, receptors, and transcription factors that allow a selective transfection and expression of opsins to modulate their activity, like the vesicular transporters that recently had been described. Herein we review the molecular targets and their colocalization with the vesicular transporter of glutamate. We will also review how optical control of specific sensory fibers could drive the nociceptive process.

## 2. Neural Types in the Dorsal Ganglia Root and Their Colocalization with the Vesicular Transporter of Glutamate

The A-β fibers have an axon diameter greater than the others, and are highly myelinated. Therefore, their velocity of conduction is between 30–80 m/s [8], and they respond to innocuous mechanical stimulation. The projection of these neurons makes synaptic contacts in the III-V layers with second order neurons in the dorsal horn of the spinal cord. Their molecular profile (Table 1, Section 1) includes the following protein markers that are characteristic of large sized neurons in the dorsal root ganglion (DRG), such as NF200 (neurofilament 200), NEFH (neurofilament heavy), Thy-1 (Thymocyte differentiation antigen 1), TrKC (Tropomyosin receptor kinase C), Cdh12 (Cadherin 12), and parvalbumin [7,9,10,11], biomarkers that are correlated with the survival and stability of nerve terminals related to dynamic touch, gentle skin stroke, kinesthetic sense of position, skin indentation, poke, pressure on guard hairs, or texture perception by fingertips [11].

The A-β fibers neurons also express specific channels, such as Nav1.1 and 1.6 (voltage-gated sodium channel), ASIC1 (acid sensing ion channel subunit 1), HCN1 (hyperpolarization activated cyclic nucleotide gated potassium channel 1) and HCN2 (hyperpolarization-activated cyclic nucleotide gated potassium channel 2) [9,12,13,14]. Although these HCN channels are expressed in a variety of sensory neuron types, their expression in A-β neurons is related to their conduction velocities and frequencies due to their current properties [15,16].

Transcription factors are another set of important markers that act as regulators of lineage-specific gene expression. Among the A-β fibers are RunX3 (runt-related transcription factor 3), ETV1 (ETS variant transcription factor 1), ETV4 (ETS variant transcription factor 4) [9,17,18,19], c-Ret and MafA/c-Maf. At the same time, three subpopulations can be differentiated, those that express Ret+ MafA+, Ret+ MAfA+ TrkC+, and MAFA+ TrkB+ Shox2+. The presence of these markers conditions the expression of TrkC, and the differentiation to A-β fibers [20,21].

In addition to the fibers that receive harmless sensory information, there are neuronal pathways that detect painful information such as A-δ and C-fibers. The A-δ fibers are lightly myelinated and have a relatively fast conduction velocity of 5–25 m/s. The diameter of Aδ-fibers is about 2–5 µm, and they are responsive to short-lasting and pricking nociceptive behavior, transmitting the immediate response to pain stimuli, which generates a withdrawal reaction from the source of stimulation. The markers are difficult to distinguish in A-δ neurons, since they express overlapped markers with other subtypes of proteins, channels, and metabotropic receptors. The A-δ and A-β fibers share markers, such as NF200 and NEFH, but also have proteins on their surface, such as substance P and CGRP, that are also shared in C fibers [7]. One of the markers by which they can be differentiated is the presence of the TrkB receptor and the voltage-dependent calcium channel 3.2 (Cav 3.2) [22]. Other notable differences are the presence of transcription factors such as contactin associated protein 2 (CNTNAP2) and family with sequence similarity 19 member A1 (FAM19A1) [14]. 

The C-fibers have an important role in the pathophysiology of pain, transmitting pain signals from the periphery to the central nervous system [23]. C-fibers are unmyelinated with less than 2 µm in diameter and have a relatively slow conduction velocity of approximately 0.5–2 m/s. C-fibers terminate in laminae I and II in the dorsal root ganglia, and correspond to polymodal nociceptors, which are activated by thermal, mechanical, and chemical stimuli. They have diverse markers that allow the identification of different neuronal subpopulations. The peptidergic neurons express substance P (SP) and the calcitonin-gene related protein (CGRP), while neurons that bind to isolectin B4 (IB4) are considered IB4+ neurons or non-peptidergic [14,24,25,26].

**Table 1 ijms-24-10220-t001:** Main markers of primary afferent fibers.

Fiber Type		Conduction Velocity	Vesiculator Type	Protein Markers	Channels	Membrane Receptors	Dorsal Laminar Distribution	Transcription Factor	References
Aβ		30–80 m/s	VGluT 1	NF200 Thy-1 parvalbumin Caderina SPP1 NEFH	Nav1.1 Nav1.6 HCN3 ASIC1 Nav1.8	DOR TrkC	III, IV, V	RunX3 ETV1 Etv4 Cdh12	[7,8,9,10,11,13,14,27,28,29,30,31,32,33,34,35]
Aδ		5–25 m/s	VGluT 2	NF200 NEFH CGRP Substance P	TRPM8 Nav1.8 HCN1 HCN3 Cav 3.2	TrkA TRKB MOR DOR	I, II	CNTNAP2 FAM19A RET	[7,13,14,29,31,32,33,36,37,38]
C	Peptidergic	0.5–2 m/s	VGluT 2	Substance P CGRP Cdk5 mTOR	TRPV1 TRPA1 TRPM8 Nav1.7 Nav1.8 HCN2 HCN3	D_2_ MOR MrgprA3 mGlu2 Ntrk1 PAC1 TrkA Y1R Y2R	I	cMAF GDNF HOB8 LMBX1 RET TAC1	[7,9,24,25,26,29,31,33,34,35,39,40,41,42,43,44,45,46]
	Non-peptidergic	0.5–2 m/s	VGluT 2	IB_4_ Cdk5 mTOR	TRPA1 TRPM3 TRPC3 Nav1.7 Nav1.8 Nav1.9 HCN1 HCN3 P_2_X_3_	DOR MrgprA3 MrgprB4 MrgprD Y_1_R Y_2_R	II	GfrA1 LMBX1 NGF PLXNC1 RET RunX	[7,9,13,24,26,28,29,31,33,34,36,41,44,45,46,47,48]
	LTMR	<0.5 m/s	VGluT 3	TH	TRPM8 Nav1.8 Nav1.9	TrkB	I, II, III	GfrA2 PIEZO2 RET	[14,28,29,34,36,48]

There are protein markers or transcription factors that allow the differentiation of C fibers. One important protein family is the transient receptor potential (TRP) channel, a cationic channel that senses the injurious information. There are six families of TRPs, the canonical (TRPC), vanilloid (TRPV), ankyrin (TRPA), melastatin (TRPM), polycystin (TRPP), and mucolipin (TRPML) [27]. 

C-peptidergic neurons are distinguished by the expression of TRPV1, a channel that participates in high-threshold thermal sensitivity and harmful chemical capsaicin [25,39]. These fibers also express the TRPA1 and TRPM8 receptors, which function as channels that can be activated in the presence of irritating agents, inflammatory mediators, and cold temperatures. Nonpeptidergic IB4+ type-C fibers also have TRPA1; however, these fibers are differentiated by the expression of TRPM3, which is characterized by its activation in the cold thermal stimuli (−40 °C) and TRPC3, which plays an important role in sensitization to noxious stimuli [49].

The sodium channels play an important role in the generation of excitability for the transmission of pain towards the dorsal horn of the spinal cord and are distributed within the C fibers. There are nine isoforms of this protein (Nav1.1 to Nav1.9), of which Nav1.8 is found in peptidergic and nonpeptidergic fibers, while the Nav1.7 channel is expressed in peptidergic C fibers, and Nav1.9 in IB4+ neurons [28]. 

The P_2_X family are ligand-gated ion channels mediating the actions of extracellular ATP and other nucleotides released after tissue damage. The activation of these channels contributes to the generation and maintenance of pain, due to a distribution in primary afferent neurons. A total of seven P_2_X receptors have been described so far (P_2_X_1_–P_2_X_7_), with C-nonpeptidergic neurons expressing the purinergic receptor P_2_X_3_ [50,51]. 

In addition, there are metabotropic protein receptors in C-fibers, such as opioid, dopamine (DA), the Mas-related G protein-coupled receptor (Mrg), or tropomyosin receptor kinase. Opioid receptors play an important role in modulating the nociceptive signal and are distributed in the nerve circuits of pain. The MOR receptor (µ opioid receptor) colocalizes with substance P and CGRP in peptidergic fibers, while the DOR receptor (δ opioid receptor) is located in neurons that do not express neuropeptides or TRPV1 [31,52,53,54] (Figure 1). Similarly, there is also a clear difference in expression in the subpopulation of C-fibers with the dopamine receptor. It has been suggested that presynaptic and postsynaptic effects are responsible for the antinociceptive effect of DA and are mediated mainly by D_2_-like receptors [55,56,57]. A D_2_-like receptor agonist produces antinociception to mechanonociceptive stimuli but not to thermonociceptive stimuli [58,59], denoting that the D_2_ receptor is expressed mainly in IB4+ small DRG neurons. Likewise, Y_1_R is found in peptidergic neurons. IB4+ fibers express both the Y_1_R and Y_2_R receptors.

Pituitary adenylate cyclase-activating polypeptide (PACAP) belongs to the VIP/secretin/glucagon superfamily. Three distinct G protein-coupled receptors mediate the actions of PACAP and VIP, present in the spinal dorsal horn and dorsal root ganglia, suggesting an important role of PACAP signaling systems in the modulation of spinal nociceptive transmission [60]. Its effects are mainly produced through the cAMP-protein kinase A (PKA) pathway by the PACAP type I (PAC1) receptor [41], inducing the cAMP responsive element binding protein (CREB) activation. In C-fibers, PACAP and their agonists induce phosphorylation of both extracellular signal-regulated kinases (ERK) via PKA in the dorsal horn neurons. This phenomenon is associated with spontaneous nociceptive behavior. The Jun N-Terminal kinase (JNK) is activated subsequently in astrocytes directly by PAC1 receptor signaling, or via ERK signaling pathway in the dorsal horn of the spinal cord, which are involved in the long-term maintenance of pain [61]. 

C-fibers also differ in their central terminals in the spinal cord. The peptidergic subset terminates more superficially, in lamina I and the outer part of lamina II, while the IB4+-nociceptors project mainly to inner lamina II of the dorsal horn of the spinal cord [62]. A subset of C-fiber has been identified, whose characteristic is to be a low-threshold tactile mechanoreceptor (C-LTMR). These afferents are characterized by the expression of thyroxine hydroxylase (TH) and TrkB (a receptor for both brain- derived neurotrophic factor (BDNF), neurotrophin-4 (NT4), Ret, and VGluT3 [63,64,65]. 

Different combinations of molecular markers are used to separate the variety of neuronal subtypes that process various forms of nociceptive information. Understanding the etiology of pain requires research into how these molecular markers alter along each of these routes or subcircuits. Although there is more than one marker for each type of fiber, the subpopulation is defined by a specific combination of markers.

In addition to these neurochemical markers and transcription factors, other characteristics have been found in these fibers, such as the vesicular transporter. Glutamate is considered the main excitatory amino acid within the nervous system and has been identified as an important factor in the generation of central pain sensitization [30,66]. To maintain glutamate homeostasis there are two main transporters, the excitatory amino acid transporters subtype 2 (EAAT2) and vesicular glutamate transporter (VGluT). Therefore, it is important to identify the neurons that release glutamate, especially those related to the transmission of pain. There are three VGluT subtypes that have been identified so far. The differences in vesicle subtypes have been associated with the colocalization of the primary afferent fibers. One example is the VGluT1 that is present in most of the large-sized neurons of the DRG; these neurons originate from the A-β fibers [67,68]. Moreover, it has been demonstrated that VGluT2 colocalizes with CGRP and IB4+ neurons, suggesting that C-peptidergic and C-non-peptidergic package glutamate by this subtype of vesiculator [24,67]; and C-LTMR are VGluT-3 [69]. 

## 3. Pain Pathway Alteration

One of the main signaling pathways involved in the inhibitory process of neuronal excitability is the GABAergic system. The GABA receptor is known as the main regulator of the painful signal, promoting hyperpolarization of neurons in the spinal cord [70]. The injury to the nerve promotes anatomical changes that cause a reduction in the number of inhibitory synapses in the dorsal spinal cord. In addition to this, molecular changes are generated that modify the inhibition process through the GABA receptor [71,72,73]. The excitability of CNS is regulated by intracellular Cl^-^, mainly by changes in its intracellular concentrations. These changes are sensed by the protein kinase WNK1, which will activate the SPAK kinase and subsequently control the permeability of the KCC2 and NKCC1 transporters. When these transporters are phosphorylated, the internal concentration of Cl^-^ is higher during chronic pain. This change in internal Cl^-^ concentration impairs the hyperpolarizing function of GABA, so that in the chronic stage of neuropathic pain GABA now becomes depolarizing (Figure 2). These changes in the GABAergic function modify the intrinsic excitability of projection neurons in the spinal cord [73,74]. This notion raises the question of whether this change in Cl^-^ concentrations is occurring in all or only some neurons, and is a subject of debate that should be addressed in future studies.

Specific neuronal markers, such as those found in C and Aδ fibers that transmit nociceptive signals, can be used as promoters to achieve specific expression of opsins. This is crucial to apply tools such as optogenetics and chemogenetics to selectively manipulate pain circuits and better understand the underlying mechanisms of pain, as well as develop new therapies for chronic pain treatment.

## 4. Optomodulation of Peripheral Nerve Activity

Among the problems in understanding neuronal circuits was the control of a group of neurons in a specific way without affecting neighboring cells and understanding cell physiology. This was possible by the application of optogenetics control on physiological processes. Optogenetics is a technique that involves the use of light to control neuronal activity through ion channels [76,77]. This novel method allows the activation or inactivation of excitable cells through the expression of types of optical activated channels called opsins, of which the main are channelrhodopsin (ChR2), halorhodopsin (NpHR), and archeorodopsin (ArchT). The study and understanding of nerve circuits that are related to the perception and transmission of pain toward the spinal cord have been favored by the selective modulation of neuronal subsets by optogenetics.

ChR2 is a protein that is activated by a luminous beam with a wavelength of 450–480 nm (blue light) and its stimulation allows the passage of cations such as Na^+^, K^+^, and Ca^++^ into the cell, which in the case of a neuronal cell generates a depolarization that creates an action potential (i.e., activation) [78]. The other protein, NpHR, is activated with a luminous beam with a wavelength of 570–600 nm. Its activation generates the entry of Cl^-^ to the neuron, causing a negative membrane potential that results in a phenomenon of hyperpolarization, having the inactivation of neuron as final effect [79]. A similar effect is observed with the expression of ArchT channel; however, this hyperpolarization is given by the H+ output. In the case of this protein, the activation is also given by the exposure to yellow light with a wavelength of 520–560 nm [80]. The expression of these proteins can be achieved by viral transfection or breeding strategies through molecular markers, which will allow the control of the neuronal activity in a specific way.

One of the first methods used is through non-transgenic (i.e., wild-type) animals and the use of viral vectors that are not Cre-dependent. This is achieved by endowing the virus with the opsin gene together with a non-specific promoter, such as the human synapsin 1 gene (hsyn), when the aim is to perform a generalized (i.e., non-specific) neuronal transfection. Other widely used non-specific promoters are CaMKII, which directs transfection mainly towards glutamatergic neurons, or VGAT, for GABAergic neurons [81]. The use of specific promoters provide specificity to the cell and as a consequence, to the model.

The other method for transfecting viral vectors is by using transgenic animals, such as those with Cre, Dre, or FLP recombinase systems. These recombinases are enzymes that catalyze recombination between two relatively close sites and the genetic material between them. When a viral vector containing the opsin gene between the recognition sites is introduced in animals whose cells have the recombinase driven by a specific promoter, they will express the opsin in specific neurons of interest [82]. Nevertheless, in order to be specific for the neuronal subtype, it is necessary to use more than one promoter with at least two recombinases; this is the so-called intersection technique (i.e., VGluT2::Cre intersection with TH Flp). However, to date this intersection technique has not been used in nociceptive pathways and the published research until now still uses only one promoter.

### Optomodulation of Primary Afferent Fibers with Viral Transfections and Constitutive Expression

There are studies that have used these strategies to demonstrate the modulation of sensory pathways [22,26,53,83], which are summarized in Table 2, Table 3, Table 4 and Table 5. One of the first transfection attempts was using the Nav1.8 promoter, which is present in all sensory pathways. Therefore, this strategy was not specific to a fiber subtype, using a constitutive strategy composed by a homozygous Nav1.8—Cre mice crossed with a heterozygous Ai32 mice, which carry the ChR2(H134R)–EYFP gene in their Gt(ROSA)26Sor locus receiving Nav1.8-ChR2 (Table 2). The selective stimulation with blue light in freely moving mice resulted in them exhibiting nocifensive behaviors, paw withdrawal, paw licking, jumping, and audible vocalization. This wide variety of behaviors are explained by the global activation of nociceptive fibers, which present the expression of the Nav1.8 channel in both peptidergic and non-peptidergic neurons [83]. This work showed that the sensitization is evocated in similar way to the painful stimuli when ChR2 activates the Nav1.8+ neurons, which is present in a generalized way in afferent sensory fibers.

On the other hand, in Nav1.8-Arch mice, obtained with heterozygous Nav1.8-Cre mice crossed with homozygous Ai35 mice carrying the floxed stop-Arch-EGFP gene in the ROSA26, the mechanical and thermal sensitization was completely prevented with the stimulation of yellow light (Table 3). Additionally, optical stimulation reduced capsaicin- and zymosan-induced mechanical allodynia [39]. These results together suggest that afferent fibers with Nav1.8+ play an important role in the onset and maintenance of pain. These results are expected; however, it is difficult to attribute the effects to a neuronal subtype involved in pain transmission, since Nav 1.8 has a widespread distribution in primary afferent fibers.

To transfect peptidergic fibers, the Cre-dependent strategy has been used under TRPV1 promoter control. Beaudry [26] transfected a neuronal subgroup using homozygous TRPV1-Cre mice injected intrathecally with an adeno-associated virus (AAV) 2/8 virus (AAV2/8-CAG-floxed stop-ChR2[H134R]-tdTomato-WPRE) to generate TRPV1-ChR2 mice (Table 2). The behavior was reported as paw withdrawal, paw lifting, and paw licking [26]. In the same group in neurons but in constitutive strategy, cross-breeding heterozygous rosa-CAG-LSL-hChR2(H134R)-tdTomato-WPRE (Ai27D) mice with heterozygous mice with Cre recombinase inserted downstream to the TRPV1 specific expression of ChR2 in the TRPV1-Cre gene showed no place preference in the absence of photostimulation, whereas its activation with light blue triggered marked aversion of the stimulation-paired area, with a corresponding increase in time spent in the non-stimulation-paired area [84]. In opposite way, the inhibitions of TRPV1 by ArchT with an injection in the DRG of adeno-associated virus (AAV5-TRPV1-ArchT-eGFP) increase mechanical paw withdrawal threshold and the latency to thermal stimulation [22]. Thus, the modulation of c-peptidergic fibers (TRPV1+) produces burning pain or itch that generates aversion.

To activate non-peptidergic fibers, Beudry et al. [26] used the Mas-related G protein–coupled receptor subtype D (MrgD) with homozygous MrgD::CreERT2 mice crossed with homozygous Ai32 carrying in the ROSA26 locus the floxed stop-ChR2(H134R)-EYFP construct mice to generate a MrgD-ChR2 heterozygous transgenic mouse. To induce opsin expression in MrgD-ChR2 mice with Cre-inducible, tamoxifen was injected intraperitoneally. The optical stimulation with blue light evoked paw withdrawal and paw lifting mainly and lesser paw licking [26]. Overall, TRPV1 and MRGD fiber activation produced similar conduct. However, in quantitative analysis, TRPV1-ChR2 mice spent more time licking while MRGD-ChR2 mice spent more time lifting, reflecting different somatosensory perception differences that agree with the evidence of the participation of peptidergic fibers in thermoception and burning sensations and non-peptidergic in mechanical nociception [87].

A second clearly defined methodology to generate the expression of opsins that can be controlled by light in non-transgenic mice with no CRE dependent virus is the intraneural viral injection (adeno-associated virus serotype-6) expressing the step-function inhibitory ChR2 used by Iyer et al. [88], a variant of the opsins that allows neurons to be activated with less frequent light pulses, thereby reducing the exposure traditionally required. Using SwiChR2, a variant that is permeable to Cl^-^ or the chloride-conducting inhibitory channelrhodopsin (iC1C2) by AAV6-hSyn-biC1C2-TS-eYFP intraneural injection to generate hyperpolarization, the blue light stimulation produced large increases in mechanical withdrawal thresholds and thermal latency measures, because all fibers were transfected but mainly unmyelinated despite using a non-specific promoter such as hsyn-I (Table 4). In the evaluation of the long-term inhibition, the sensitivity of mechanical withdrawal threshold and thermal latency decreased, and mice exhibited ‘post-light’ inhibition property (inhibition after light stimulus has disappeared). Furthermore, when the expression of SwiChR was evaluated in sensitized mice through the injection of formalin and subsequently stimulated with blue light on the plantar paw, a reduction in behavior was observed in phase 1, but not in phase II in formalin model [88].

To transfect halorhodopsin in unmyelinated cells of the primary afferents, an injection of AAV6-hSyn-eNpHR3.0-eYFP into the sciatic nerve was used to generate halorhodopsin in unmyelinated fibers. When mechanical sensitivity was evaluated in mice stimulation with yellow light, an inhibition of nociceptors was observed through an increase in the response threshold with stimulation with the Von Frey filament and an increase in latency to thermal stimulation. If the effect is analyzed in mice with neuropathic pain symptoms generated by the CCI in mice expressing halorhodopsin model, the optogenetic inhibition in these same fibers evokes a reversal of mechanical allodynia and an increase in thermal threshold [89]. With these results it is demonstrated that light delivered transdermally raises the hyperpolarization of unmyelinated fibers and decreases the activity of nociception.

**Table 4 ijms-24-10220-t004:** Modulation of sensory pathways with optogenetics and chemogenetics using no Cre-dependent strategy in no sensitized condition.

Specificity Strategy	Construct	Fiber and Cell Type	Injection Site	Place of Stimulation	Behavioral Phenotype	Painful Condition	Reference
No Cre dependent	AAV6-hSyn-ChR2-eYFP Product: Chr2-eYFP	C-fiber: -Unmyelinated primary afferent	intrasciatic	The plantar surface Blue light (473 nm)	Decreases in paw withdrawal Place aversion	No sensitized	[89]
	AAV6-**hSyn**-SwiChR-eYFP Product: SwiChR-eYFP	C-fiber: -Unmyelinated primary afferent	intraneural	The plantar surface of the hindpaw Blue light (473 nm)	Increases in mechanical withdrawal threshold and thermal withdrawal latency		[88]
	AAV6-**hSyn**-biC1C2-TS-eYFP Product: iC1C2-eYFP	C-fiber: -Unmyelinated primary afferent	intraneural	The plantar surface of the hindpaw Blue light (473 nm)	Increases in mechanical withdrawal threshold and thermal withdrawal latency		[88]
	AAV6-**hSyn**-eNpHR3.0-eYFP	C-fiber: -Unmyelinated primary afferent	Intrasciatic	The plantar surface Yellow light (590 nm)	Increases in mechanical withdrawal threshold and thermal withdrawal latency		[89]
	AAV6-**hSyn**-HA-HM4D(Gi)-IRES-mCitrine Product:hM4D-Gi	C-fiber: -Unmyelinated primary afferent	Intraneural	Clozapine-N-oxide intraperitoneally	Increases in mechanical withdrawal threshold and thermal withdrawal latency		[88]

AHTMR-neurons, known as high threshold mechanoreceptors nociceptors, are responsible for transmitting the initial signals to the spinal cord for pain perception. When optogenetically manipulated through the injection of (AAV8)/CAG-ArchT-GFP in spinal nerves L4-L5, it generates an increase in mechanical response thresholds in Sprague-Dawley rats with neuropathic pain through the SNL model (Table 5), suggesting that this neuronal subpopulation contributes to the transmission of pain after damage to the nervous system [90]. Although the promoter and AAV8 are thought to be nonselective, the expression of ArchT was preferential in fast-conducting high-threshold mechanoreceptors (AHTMR), mainly in neurons NF200+. Possibly this restriction of expression is due to the presence of specific glycoproteins on myelinated sensory neurons that allow AAV8 greater binding and possibly improved access to the cell.

**Table 5 ijms-24-10220-t005:** Modulation of sensory pathways with optogenetic and using no Cre-dependent strategy and chemogenetics in sensitized condition.

Specificity Strategy	Construct	Fiber and Cell Type	Injection Site	Place of Stimulation	Behavioral Phenotype	Painful Condition		Reference
No Cre-dependent	AAV6-hSyn-SwiChR-eYFP Product: SwiChR-eYFP	C-fiber: Unmyelinated primary afferent	intrasciatic	The plantar surface of the hindpaw Blue light (473 nm)	Reduced pain behavior in phase I	Sensitized	Formalin test	[88]
	AAV6-**hSyn**-eNpHR3.0-eYFP Product: eNpHR-eYFP	C-fiber unmyelinated primary afferent	intraneural	The plantar surface of the hindpaw Yellow light (490 nm)	Reduction of mechanical allodynia and thermal hyperalgesia		CCI	[89]
	AAV8/**CAG**-ArchT-GFP In myelinated neurons. Product: ArchT-GFP	AHTMR-fiber myelinated primary afferent	intrathecal	The plantar surface of the hindpaw Green light	Increase in paw withdrawal threshold		SNL	[90]

CCI: Chronic Constriction Injury; SNL: Spinal Nerve Ligation.

## 5. Long-Term Chemogenic Modulation of Pain

Another strategy to modulate neural activity is through of chemogenetics. This technique is used to control neuronal activity in specific cells in a temporal and reversible manner, using a chemical molecule instead of light as in optogenetics. This technique is based on the expression of a G protein-coupled receptor (GPCR) designed to respond only to an exogenous molecule called a designer receptor, exclusively activated by designer drugs (i.e., DREADDs). One of the applications is the selective modulation using specific ligands (Salvinorin B or clozapine-N-Oxide) [91]. There are DREADDs that can modulate the neural activity through G protein coupling, either to activate by using Gq/Gs, or to inactivate by using Gi proteins.

Iyer et al. [88] carried out a chemogenetic modulation on unmyelinated peripheral fibers with injection of AAV6-hSyn-HA-hM4D(Gi)-IRES-mCitrine and subsequent intraperitoneal administration of clozapine-N-oxide, increasing mechanical withdrawal thresholds and thermal sensation in mice expressing the hM4D receptor (Table 4). The authors reported the expression of mCitrine in small-diameter neurons despite using the non-specific hsyn promoter.

As stated previously, the neurons involved in the transmission of pain from the periphery to the spinal cord are VGluT2. The approaches to modulate the activity of this neuronal subtype have been carried out by the generation of a transgenic model by cross breeding VlguT2::Cre mice with tdTomatoFlox/flox mice (VGluT2::Cre) and for chemogenetic modulation, with the injection of the viral construct AAV8- hSyn-DIO-hM3Dq-mCherry or AAV8-hSyn-DIO-HA-KORD-IRES-mCitrine in spinal dorsal horn neurons [85]. The activation of the Gq-coupled human M3 muscarinic receptor DREADD (hM3D) with intraperitoneally administration of clozapine-N-Oxide activates VGluT2-expressing glutamatergic dorsal horn neurons reduces the mechanical and thermal withdrawal threshold eliciting spontaneous flinching and licking (Table 3). Therefore, there is an increase in nociceptive activity. Moreover, when evaluated in a neuropathic pain condition generated by spared nerve injury, this caused a reduction of mechanical withdrawal threshold induced by an increase in glutamatergic input to VGluT2 neurons; however, Salbinorin B (an agonist for Kappa-opioid-receptor) evokes a wide reduction in the mechanical response threshold in VGluT2::Cre mice injected with AAV-hSyn-DIO-HA-KORD-IRES-mCitrine with neuropathic condition [85]. This indicates that VGluT2-expressing dorsal horn neurons are critically involved in maintaining pain hypersensitivity.

Optogenetics and chemogenetics have been used at the preclinical level to study the physiology of the peripheral nervous system and its relationship to pain processing [88,92,93]. Initial studies using optogenetics to stimulate large populations of sensory neurons showed that the stimulation of these neurons induced pain-like behaviors [83,89]. Conversely, transfecting and activating photosensitive or chemosensitive proteins that inhibit neuronal activity can also be effective in reducing sensitivity to pain in both healthy and inflamed or neuropathic animals [22,26,88,89]. These studies provide preliminary evidence that viral delivery of inhibitory photosensitive proteins could be a non-pharmacological way to treat chronic pain in humans in the future.

Currently, pharmacological therapy is used to treat different pain conditions, but these strategies have limitations, such as lack of specificity and control over the timing of treatments. Optogenetics offers a way to improve both target specificity and temporal control of pain management in patients with chronic pain. However, there are notable hurdles to overcome in the development of safe and effective gene therapy vectors and biocompatible light delivery systems to provide control over the location, duration, and intensity of opsin activation. 

Optogenetics has potential clinical applications beyond silencing nociceptive neurons. It has been used to treat retinitis pigments, which has been shown to partially restore vision in a photoreceptor. The U.S. Food and Drug Administration has approved an AAV-mediated gene therapy to treat an inherited retinal disease [94,95]. This suggests a promising use of these tools in the future against other pathologies, including chronic pain.

Despite optogenetics being a promising technique for chronic pain treatment, there are still limitations that need to be addressed for its successful clinical use. Some of the main limitations include virus delivery, which requires access to the nerve or spinal cord through blood or cerebrospinal fluid, which is limited and invasive. Another important point is the activation of opsin, which requires the entry of light at a specific wavelength. Precise and efficient light control devices are needed to achieve optimal activation of photosensitive proteins. Additionally, the subject’s immune response must be considered. Immunogenicity is a common problem associated with viral delivery. The immune system in patients may recognize the virus or photosensitive protein as foreign and mount an immune response, which can reduce the effectiveness of the treatment. Furthermore, the duration of treatment must be considered. Optogenetics is a relatively new technique, and more research is needed to determine the optimal duration of treatment and how the technique can be maintained in the long term. Finally, it is important to note that there are high Cl^-^ concentrations in neurons that have an increment in the excitability from the nociceptive pathway [75], which can limit the effectiveness of optogenetic chloride pumps.

## 6. Conclusions

Optogenetics and chemogenetics are powerful techniques that allow for selective manipulation of neuronal activity and could be used as a strategy for modulating pain. However, identifying which neuronal subtypes to manipulate depending on the pain condition may be complex, which is why the classification and identification of sensory fibers is important. It is well-described that there are specialized mechanisms that trigger the appearance of painful behaviors. An important question yet to be answered is what neuronal circuit participates in either allodynia or hyperalgesia. To date, there are no studies using specific promoters directed to a particular neuronal population that could be important to delineate the pathophysiological processes mediating chronic neuropathic pain. 

## Figures and Tables

**Figure 1 ijms-24-10220-f001:**
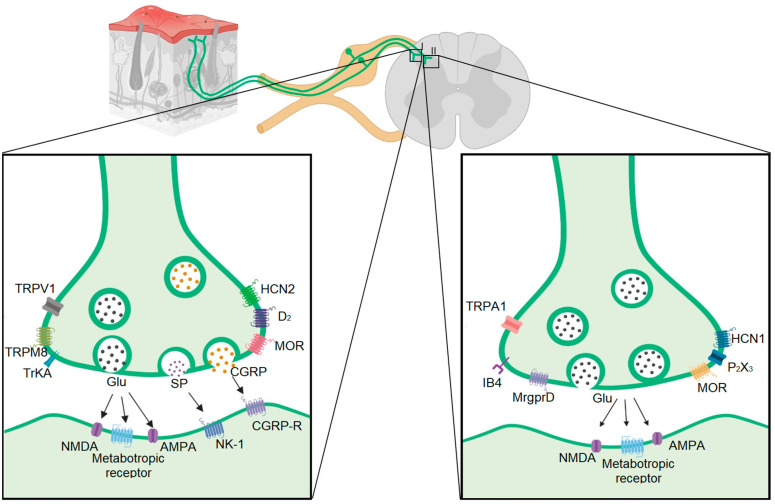
Main proteins expressed in peptidergic (**left side**) and non-peptidergic (**right side**) VGluT2 neurons. There are molecular markers for their differentiation as CGRP, SP, and TRPV1 in peptidergic neurons, and on the other hand MrgprD and IB4 in non-peptidergic. (Image developed from Table 1, C-fiber section).

**Figure 2 ijms-24-10220-f002:**
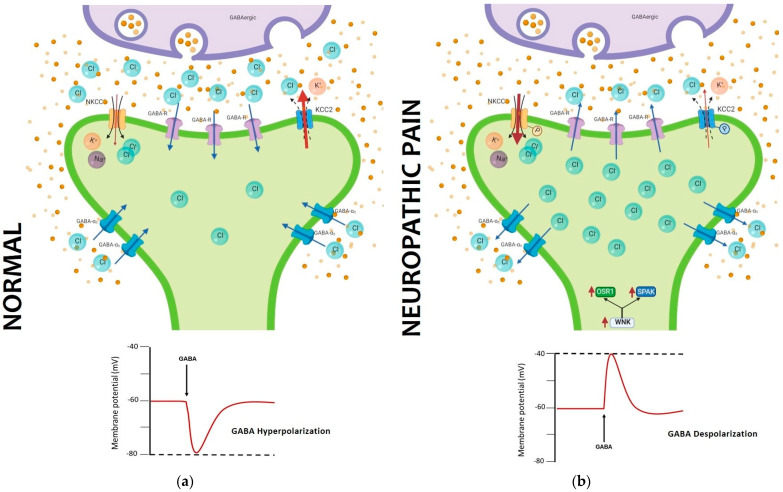
GABA impairment in neuropathic pain is caused by a shift in chloride concentration. (**a**) Normal NKCC1 and KCC2 activity maintains a lower concentration inside, allowing chloride to enter the cell and be hyperpolarizing when GABA is activated; (**b**) in neuropathic pain, phosphorylation of NKCC1 and KCC2 inverts the concentration of chloride, and when GABA is activated, there is an efflux of chloride depolarizing the neuron. (Image developed from [73,74,75]).

**Table 2 ijms-24-10220-t002:** Modulation of sensory pathways with optogenetic using Cre-dependent strategy in no sensitized condition.

Specificity Strategy	Construct	Fiber and Cell Type	Injection Site	Place of Stimulation	Behavioral Phenotype	Painful Condition	Reference
**Cre dependent**	**Nav1.8**: Cre/Ai32 (carry the ChR2(H134R)–EYFP in Gt(ROSA)26Sor locus) Product: **Nav1.8**–ChR2+	Aβ Fiber Aδ Fiber C-fibers: -Peptidergic and -nonpeptidergic	NA	The plantar surface of the hindpaw blue light (473 nm)	Paw withdrawal and paw licking	Not sensitized	[83]
	**TRPV1**: Cre/AAV5-TRPV1-ArchT-eGFP Product: **TRPV1**-Arch+	C-fiber Peptidergic	DRG injection	The plantar surface of the hindpaw Green light (532 nm)	Increases in mechanical withdrawal threshold and thermal latency		[22]
	**TRPV1**: Cre/AAV2/8-CAG-floxed stop-ChR2[H134R]-tdTomato-WPRE Product: **TRPV1**-ChR2	C-fiber Peptidergic	Intrathecal	The plantar surface of the hindpaw Blue light (473 nm)	Increases in paw withdrawal, paw lifting and paw licking		[26]
	**TRPV1**: Cre/Ai27D (carry Rosa-CAG-LSL-hChR2(H134R)-tdTomato-WPRE) Product: **TRPV1**-ChR2	C-fiber Peptidergic	NA	Epineural in sciatic nerve Blue light (470 nm)	Thermal and mechanical sensitivity		[84]
	**MrgD**: CreERT2/Ai32 (carry in the ROSA26 locus the floxed stop-ChR2(H134R)-EYFP) Product: **MrgD**-ChR2 (Opsin induced by tamoxiofen)	C-fiber Nonpeptidergic	NA	The plantar surface of the hindpaw Blue light (473 nm)	Paw withdrawal and lifting		[26]
	**VGluT2**: Cre/AAV8-hSyn-DIO-hM3Dq-mCherry Product: VGluT2- hM3D-Gq	C-fiber: -Peptidergic and –nonpeptidergic (VGluT2-dorsal horn)	Dorsal horn of the spinal cord	clozapine N-oxide intraperitoneally	Increases in mechanical and thermal sensitivities		[85]

NA: Not apply.

**Table 3 ijms-24-10220-t003:** Modulation of sensory pathways with optogenetic using Cre-dependent strategy in sensitized condition.

Specificity Strategy	Construct	Fiber and Cell Type	Injection Site	Place of Stimulation	Behavioral Phenotype	Painful Condition		Reference
Cre-dependent	**VGluT3**: Cre/Ai32 carry ChR2 Product: **VGluT3**-ChR2	LTMR Fiber: VGluT3+	NA	The plantar surface of the hindpaw Blue light (470 nm)	Elicit nociceptive behavior	sensitized	Oxaliplatin-induced neuropathy	[48]
	**Nav1.8**: Cre/Ai35 (Carry the floxed stop-Arch-EGFP gene in the ROSA26 locus) Product: **Nav1.8**-Arch+	C-fiber Peptidergic	NA	The plantar surface of the hindpaw Yellow light	Decrease in mechanical allodynia		Spared nerve injury	[39]
	**VGluT2**: Cre/AAV8-hSyn-DIO-HA-KORD-IRES-mCitrine Product: **VGluT2**-KORD-Gi	Peptidergic and nonpeptidergic (VGluT2-dorsal horn)	Dorsal horn of the spinal cord	Salvinorin B intraperitoneal	Decrease in tactile allodynia/ Decrease in pain hypersensibility		SNI/CFA	[85]
	**Thy1**-COP4/YFP/W-TChR2V4	Aβ fibers	NA	The plantar surface of the hindpaw Yellow light	Lifting and flinching behaviors		PNI	[86]

NA: Not apply; SNI: Spared Nerve Injure; CFA: The Complete Freund’s Adjuvant.

## Data Availability

Not applicable.

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
