# Peer review of "Optogenetic and Chemogenic Control of Pain Signaling: Molecular Markers"

_ijms, 2023, doi:10.3390/ijms241210220_

Round 1

Reviewer 1 Report

This manuscript concerns about optogenetic and chemogenic control of pain signaling. This review addresses the molecular markers  present in the nociceptive fibers responsible for sensory signals as use as optogenetics targets to modulate nociceptive behavior.

This work promise finding adequate and promising  molecular markers. Optogenetics is a promising tool.

Abstract – Rewrite it. It is not concerned about point of review.

The first paragraph in the introduction  is without references. Add them.

Paragraph: „With recent techniques as optogenetics or chemogenetics is possible to activate or inactivate neuronal circuits in a specific way; thus, is essential to identify molecular targets from each fiber, such as channels, receptors and transcription factors that allow a  selective transfection and expression of opsins to modulate their activity, like the vesicular transporters that recently had been described.“  Without reference- add them

The C-fibers have an important role in the pathophysiology of pain transmit, [20]. – Finish this sentence.

Table 1. Main markers of primary afferent fibers. – This table is interesting and clear. It is presented in the good way markers of afferent fibers.

Figure1  - Is this figure made by the authors or taken from other source? It has to be more clear and in the better resolution.

Table 2. Modulation of sensory pathways with optogenetic and chemogenetic. – This table is too long. Can you divide this table into few parts?

Change the conclusion.

It is good. Minor typographic errors.

Author Response

Point 1: This manuscript concerns about optogenetic and chemogenic control of pain signaling. This review addresses the molecular markers  present in the nociceptive fibers responsible for sensory signals as use as optogenetics targets to modulate nociceptive behavior.

This work promise finding adequate and promising  molecular markers. Optogenetics is a promising tool.

Abstract – Rewrite it. It is not concerned about point of review.

Response 1: Thank you for your observation. We have restructured the abstract.

 Point 2: The first paragraph in the introduction  is without references. Add them.

Response 2: Thank you for your comment, we have added the references.

Point 3: Paragraph: „With recent techniques as optogenetics or chemogenetics is possible to activate or inactivate neuronal circuits in a specific way; thus, is essential to identify molecular targets from each fiber, such as channels, receptors and transcription factors that allow a  selective transfection and expression of opsins to modulate their activity, like the vesicular transporters that recently had been described.“  Without reference- add them

Response 3: Thank you for your observation, we have added the reference to the text.

 Point 4: The C-fibers have an important role in the pathophysiology of pain transmit, [20]. – Finish this sentence.

Response 4: Thank you for your observation, we have rephrased the sentence to make it more understandable.

 Point 5: Table 1. Main markers of primary afferent fibers. – This table is interesting and clear. It is presented in the good way markers of afferent fibers.

Response 5: Thank you for your comment, which are very valuable to us.

Point 6: Figure1- Is this figure made by the authors or taken from other source? It has to be more clear and in the better resolution.

Response 6: Thank you for your comments. The image was created by us, and we have added this information in the figure caption. Regarding the image quality, although the original image appears to have good quality, some issues may arise when converting it to PDF. Nonetheless, we have tried to improve the resolution as much as possible.

Point 7: Table 2. Modulation of sensory pathways with optogenetic and chemogenetic. – This table is too long. Can you divide this table into few parts?

Response 7: Thank you for your suggestion. We have divided the table and created four separate tables, which are now Table 2, 3, 4, and 5.

Point 8: Change the conclusion.

Response 8: Thank you for your observation, we have restructured the conclusion.

Note: we made the changes in red in manuscript

Reviewer 2 Report

This review article addressed the molecular markers present in the nociceptive fibers responsible for sensory signals. Please conduct the concerns below.

1.      Example of the markers must indicate in the abstract.

2.      Modulation of sensory pathways with optogenetic and chemogenetic that needs to introduce in detail.

3.      Progress of the functional pain was not conducted in clear.

4.      In chronic neuropathic pain, an increase in nociceptive activity has been established due to multiple parameters. A chemogenetic modulation of pain has been mentioned but it seems not clear in detail.

5.      Optogenetics is suggested as a promising tool in conclusion. Real example with reference(s) seems required to support it.

6.      Limitation(s) of current review will be helpful.

This is an intriguing review piece. The authors clearly understood the context. However, it appears difficult for junior researchers to follow. As a result, I recommend shortening and clarifying the language. Thank you kindly.

Author Response

Point 1: This review article addressed the molecular markers present in the nociceptive fibers responsible for sensory signals. Please conduct the concerns below.

  1. Example of the markers must indicate in the abstract.

Response 1: Thank you for your observation, we have restructured the abstract following your comment and added examples of markers.

 Pont 2:  Modulation of sensory pathways with optogenetic and chemogenetic that needs to introduce in detail.

Response 2: Thank you for ypur comment, We have added a text before point 4 "Optomodulation of peripheral nerve activity" to introduce it.

Point 3:    Progress of the functional pain was not conducted in clear.

Response 3. Thank you for your comment. We have restructured part of the manuscript where pain physiology is discussed in an attempt to make it more comprehensible

 Point 4: In chronic neuropathic pain, an increase in nociceptive activity has been established due to multiple parameters. A chemogenetic modulation of pain has been mentioned but it seems not clear in detail.

Response 4. Thank you for your observation. We have rewritten part of the description addressing the evaluation of chemogenetics in neuropathic pain, which can be found between lines 385-390.

 Point 5:  Optogenetics is suggested as a promising tool in conclusion. Real example with reference(s) seems required to support it.

Response 5. Thank you for your comment. In this regard, we have generated a paragraph that highlights the promising use of optogenetics with the corresponding references. However, we believe it would be better to place it before the conclusion.

Point 6: Limitation(s) of current review will be helpful.

Response 6: We agree with your comment regarding expressing the limitations, which were added in the last paragraph before the conclusion.

Round 2

Reviewer 1 Report

I suggest to accept this manuscript, without any changes.

Author Response

Dear reviewer

We made the corrections tha the the academic editor suggest and we change also the abstract:

Abstract: Pain is a complex experience that involves physical, emotional, and cognitive aspects. Understanding the neural circuits involved in its pathophysiology is important for developing treatment strategies. Technological advancements such as optogenetics and chemogenetics have enabled researchers to selectively activate or deactivate specific neural circuits, shedding light on the key neuronal subtypes involved. In this article, molecular targets in sensory fibers are explored, including channels like TRPV1 in peptidergic C fibers, TRPA1 in non-peptidergic C fibers, differentially expressed receptors such as MOR and DOR, and transcription factors, along with their colocalization with the vesicular glutamate transporter. The aim for this characterization is to identify efficiently neurons that can be modulated using optogenetics or chemogenetics. However, a completely specific marker for each subtype of neurons has not yet been discovered. On the other hand, it is possible to identify a set or combination of markers that define a subpopulation, allowing for greater specificity in the development of these strategies.